# AN INFORMATION-THEORETIC ANALYSIS OF DEEP LATENT-VARIABLE MODELS

## ABSTRACT

We present an information-theoretic framework for understanding trade-offs in unsupervised learning of deep latent-variables models using variational inference. This framework emphasizes the need to consider latent-variable models along two dimensions: the ability to reconstruct inputs (distortion) and the communication cost (rate). We derive the optimal frontier of generative models in the two-dimensional rate-distortion plane, and show how the standard evidence lower bound objective is insufficient to select between points along this frontier. However, by performing targeted optimization to learn generative models with different rates, we are able to learn many models that can achieve similar generative performance but make vastly different trade-offs in terms of the usage of the latent variable. Through experiments on MNIST and Omniglot with a variety of architectures, we show how our framework sheds light on many recent proposed extensions to the variational autoencoder family.

## 1 INTRODUCTION

Deep learning has led to tremendous advances in supervised learning (Szegedy et al., 2017; Huang et al., 2017; Vaswani et al., 2017); however, unsupervised learning remains a challenging area. Recent advances in variational inference (VI) (Kingma & Welling, 2014; Rezende et al., 2014), have led to an explosion of research in the area of deep latent-variable models and breakthroughs in our ability to model natural high-dimensional data. This class of models typically optimize a lower bound on the log-likelihood of the data known as the evidence lower bound (ELBO), and leverage the "reparameterization trick" to make large-scale training feasible.

However, a number of papers have observed that VAEs trained with powerful decoders can learn to ignore the latent variables (Chen et al., 2017; Tomczak & Welling, 2017; Bowman et al., 2016). We demonstrate this empirically and explain the issue theoretically by deriving the ELBO in terms of the mutual information between $X$, the data, and $Z$, the latent variables. Having done so, we show that the previously-described $\beta$-VAE objective (Higgins et al., 2017) has a theoretical justification in terms of a Legendre-transformation of a constrained optimization of the mutual information. This leads to the core point of this paper, which is that the optimal *rate* of information in a model is task-dependent, and optimizing the ELBO directly makes the selection of that rate purely a function of architectural choices, whereas by using $\beta$-VAE or other constrained optimization objectives, practitioners can learn models with optimal rates for their particular task without having to do extensive architectural search.

Mutual information provides a reparameterization-independent measure of dependence between two random variables. Computing mutual information exactly in high dimensions is problematic (Paninski, 2003; Gao et al., 2017), so we turn to recently developed tools in variational inference to approximate it. We find that a natural lower and upper bound on the mutual information between the input and latent variable can be simply related to the ELBO, and understood in terms of two terms: (1) a lower bound that depends on the distortion, or how well an input can be reconstructed through the encoder and decoder, and (2) an upper bound that measures the rate, or how much information is retained about the input. Together these terms provide a unifying perspective on the set of optimal models given a dataset, and show that there exists a continuum of models that make very different trade-offs in terms of rate and distortion.

By leveraging additional information about the amount of information contained in the latent variable, we show that we can recover the ground-truth generative model used to create the data in a toy model. We perform extensive experiments on MNIST and Omniglot using a variety of encoder, decoder, and prior architectures and demonstrate how our framework provides a simple and intuitive mechanism for understanding the trade-offs made by these models. We further show that we can control this tradeoff directly by optimizing the $\beta$-VAE objective, rather than the ELBO. By varying $\beta$, we can learn many models with the same architecture and comparable generative performance (in terms of marginal data log likelihood), but that exhibit qualitatively different behavior in terms of the usage of the latent variable and variability of the decoder.

## 2 FRAMEWORK

**Unsupervised Representation Learning**   Depending on the task, there are many desiderata for a good representation. Here we focus on one aspect of a learned representation: the amount of information that the latent variable contains about the input. In the absence of additional knowledge of a "downstream" task, we focus on the ability to recover or reconstruct the input from the representation. Given a set of samples from a true data distribution $p^*(x)$, our goal is to learn a representation that contains a particular amount of information and from which the input can be reconstructed as well as possible.

We will convert each observed data vector $x$ into a latent representation $z$ using any stochastic encoder $e(z|x)$ of our choosing. This then induces the joint distribution $p_e(x, z) = p^*(x)e(z|x)$ and the corresponding marginal $p_e(z) = \int dx\, p^*(x)e(z|x)$ (the "aggregated posterior" in Makhzani et al. (2016); Tomczak & Welling (2017)) and conditional $p_e(x|z) = p_e(x, z)/p_e(z)$.

A good representation $Z$ must contain information about the input $X$ which we define as follows:

$$\mathrm{I_{rep}}(X; Z) = \iint dx\, dz\, p_e(x, z) \log \frac{p_e(x, z)}{p^*(x)p_e(z)}. \tag{1}$$

We will call this the *representational mutual information*, to distinguish it from the *generative mutual information* we discuss in Appendix C. Equation 1 is hard to compute, since we do not have access to the true data density $p^*(x)$, and computing the marginal $p_e(z) = \int dx\, p_e(x, z)$ can be challenging. As demonstrated in Barber & Agakov (2003); Agakov (2006); Alemi et al. (2017) there exist useful, tractable variational bounds on mutual information. The detailed derivation for this case is included in Appendices B.1 and B.2 for completeness. These yield the following lower and upper bounds:

$$\int dx\, p^*(x) \int dz\, e(z|x) \log \frac{d(x|z)}{p^*(x)} \leq \mathrm{I_{rep}}(X; Z) \leq \underbrace{\int dx\, p^*(x) \int dz\, e(z|x) \log \frac{e(z|x)}{m(z)}}_{\text{rate}(R)} \tag{2}$$

where $d(x|z)$ (the "*decoder*") is a variational approximation to $p_e(x|z)$, $m(z)$ (the "*marginal*") is a variational approximation to $p_e(z)$, and all the integrals can be approximated using Monte Carlo given a finite sample of data from $p^*(x)$, as we discuss below.

In connection with rate-distortion theory, we can interpret the upper bound as the *rate R* of our representation (Tishby & Zaslavsky, 2015). This rate term measures the average number of additional nats necessary to encode samples from the encoder using an entropic code constructed from the marginal, being an average KL divergence. Unlike most rate-distortion work (Cover & Thomas, 2012), where the marginal is assumed a fixed property of the channel, here the marginal is a completely general distribution, which we assume is learnable. Similarly, we can interpret the lower bound as the *data entropy H*, which measures the complexity of the dataset (a fixed but unknown constant), minus the *distortion D*, which measures our ability to accurately reconstruct samples:

$$\underbrace{\left( -\int dx\, p^*(x) \log p^*(x) \right)}_{\text{data entropy}(H)} - \underbrace{\left( -\int dx\, p^*(x) \int dz\, e(z|x) \log d(x|z) \right)}_{\text{distortion}(D)} \leq \mathrm{I_{rep}} . \tag{3}$$

This distortion term is defined in terms of an arbitrary *decoding* distribution $d(x|z)$, which we consider a learnable distribution. This contrasts with most of the compression literature where distortion is typically measured using a fixed perceptual loss (Ballé et al., 2017). Combining these equations, we get the "sandwich equation" $H - D \leq \mathrm{I_{rep}} \leq R$. Notice that ELBO $= -D - R$.

**Phase Diagram** From the sandwich equation, we see that $H - D - R \leq 0$. This is a bound that must hold for any set of four distributions $p^*(x), e(z|x), d(x|z), m(z)$. The inequality places strict limits on which values of rate and distortion are achievable, and allows us to reason about all possible solutions in a two dimensional $RD$-plane. A sketch of this phase diagram is shown in Figure 1.

First, we consider the data entropy term. For discrete data[1], all probabilities in $X$ are bounded above by one and both the data entropy and distortion are non-negative ($H \geq 0, D \geq 0$). The rate is also non-negative ($R \geq 0$), because it is an average KL divergence, for either continuous or discrete $Z$. The positivity constraints and the sandwich equation separate the $RD$-plane into feasible and infeasible regions, visualized in Figure 1. The boundary between these regions is a convex curve (thick black line). Even given complete freedom in specifying the encoder $e(z|x)$, decoder $d(x|z)$ and marginal approximation $m(z)$, and infinite data, we can never cross this bounding line.

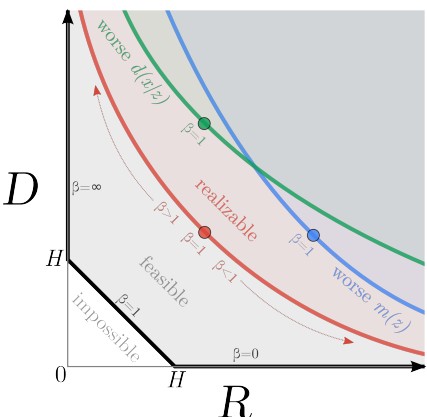

Figure 1: Schematic representation of the phase diagram in the $RD$-plane. The *distortion* ($D$) axis measures the reconstruction error of the samples in the training set. The *rate* ($R$) axis measures the relative KL divergence between the encoder and our own marginal approximation. The thick black lines denote the feasible boundary in the infinite model capacity limit.

We now explain qualitatively what the different areas of this diagram correspond to. For simplicity, we will consider the infinite model family limit, where we have complete freedom in specifying $e(z|x), d(x|z)$ and $m(z)$ but consider the data distribution $p^*(x)$ fixed.

The bottom horizontal line corresponds to the zero distortion setting, which implies that we can perfectly encode and decode our data; we call this the *auto-encoding limit*. The lowest possible rate is given by $H$, the entropy of the data. This corresponds to the point ($R = H, D = 0$). (In this case, our lower bound is tight, and hence $d(x|z) = p_e(x|z)$.) We can obtain higher rates at fixed distortion by making the marginal approximation $m(z)$ a weaker approximation to $p_e(z)$, since only the rate and not the distortion depends on $m(z)$.

The left vertical line corresponds to the zero rate setting. Since $R = 0 \implies e(z|x) = m(z)$, we see that our encoding distribution $e(z|x)$ must itself be independent of $x$. Thus the latent representation is not encoding any information about the input and we have failed to create a useful learned representation. However, by using a suitably powerful decoder, $d(x|z)$, that is able to capture correlations between the components of $x$ (e.g., an autoregressive model, such as pixelCNN (Salimans et al., 2017), or an acausal MRF model, such as (Dai et al., 2015)), we can still reduce the distortion to the lower bound of $H$, thus achieving the point ($R = 0, D = H$); we call this the *auto-decoding limit*. Hence we see that we can do density estimation without learning a good representation, as we will verify empirically in Section 4. (Note that since $R$ is an upper bound on the mutual information, in the limit that $R = 0$, the bound must be tight, which guarantees that $m(z) = p_e(z)$.) We can achieve solutions further up on the $D$-axis, while keeping the rate fixed, simply by making the decoder worse, since only the distortion and not the rate depends on $d(x|z)$.

Finally, we discuss solutions along the diagonal line. Such points satisfy $D = H - R$, and hence both of our bounds are tight, so $m(z) = p_e(z)$ and $d(x|z) = p_e(x|z)$. (Proofs of these claims are given in Sections B.3 and B.4 respectively.)

So far, we have considered the infinite model family limit. If we have only finite parametric families for each of $d(x|z), m(z), e(z|x)$, we expect in general that our bounds will not be tight. Any failure

---

[1] If the input space is continuous, we can consider an arbitrarily fine discretization of the input.

of the approximate marginal $m(z)$ to model the true marginal $p_e(z)$, or the decoder $d(x|z)$ to model the true likelihood $p_e(x|z)$, will lead to a gap with respect to the optimal black surface. However, it will still be the case that $H - D - R \leq 0$. This suggests that there will still be a one dimensional optimal surface, $D(R)$, or $R(D)$ where optimality is defined to be the tightest achievable sandwiched bound within the parametric family. We will use the term $RD$ *curve* to refer to this optimal surface in the rate-distortion ($RD$) plane. Since the data entropy $H$ is outside our control, this surface can be found by means of constrained optimization, either minimizing the distortion at some fixed rate, or minimizing the rate at some fixed distortion, as we show below. Furthermore, by the same arguments as above, this surface should be monotonic in both $R$ and $D$, since for any solution, with only very mild assumptions on the form of the parametric families, we should always be able to make $m(z)$ less accurate in order to increase the rate at fixed distortion (see shift from red curve to blue curve in fig. 1), or make the decoder $d(x|z)$ less accurate to increase the distortion at fixed rate (see shift from red curve to green curve in fig. 1).

**Optimization**   In this section, we discuss how we can find models that target a given point on the $RD$ curve. Recall that the rate $R$ and distortion $D$ are given by

$$R \equiv \int dx\, p^*(x) \int dz\, e(z|x) \log \frac{e(z|x)}{m(z)} \tag{2}$$

$$D \equiv -\int dx\, p^*(x) \int dz\, e(z|x) \log d(x|z) \tag{3}$$

These can both be approximated using a Monte Carlo sample from our training set. We also require that the terms $\log d(x|z)$, $\log m(z)$ and $\log e(z|x)$ be efficient to compute, and that $e(z|x)$ be efficient to sample from. In Section 4, we will describe the modeling choices we made for our experiments.

In order to explore the qualitatively different optimal solutions along the frontier, we need to explore different rate-distortion trade-offs. One way to do this would be to perform some form of constrained optimization at fixed rate. Alternatively, instead of considering the rate as fixed, and tracing out the optimal distortion as a function of the rate $D(R)$, we can perform the Legendre transformation and can find the optimal rate and distortion for a fixed $\beta = \frac{\partial D}{\partial R}$, by minimizing $\min_{e(z|x),m(z),d(x|z)} D + \beta R$. Writing this objective out in full, we get

$$\min_{e(z|x),m(z),d(x|z)} \int dx\, p^*(x) \int dz\, e(z|x) \left[ -\log d(x|z) + \beta \log \frac{e(z|x)}{m(z)} \right]. \tag{4}$$

If we set $\beta = 1$, this matches the ELBO objective used when training a VAE (Kingma & Welling, 2014), with the distortion term matching the reconstruction loss, and the rate term matching the "KL term". Note, however, that this objective does not distinguish between any of the points along the diagonal of the optimal $RD$ curve, all of which have $\beta = 1$ and the same ELBO. Thus the ELBO objective alone (and the marginal likelihood) cannot distinguish between models that make no use of the latent variable (autodecoders) versus models that make large use of the latent variable and learn useful representations for reconstruction (autoencoders). This is demonstrated experimentally in Section 4.

If we allow a general $\beta \geq 0$, we get the $\beta$-VAE objective used in (Higgins et al., 2017; Alemi et al., 2017). This allows us to smoothly interpolate between auto-encoding behavior ($\beta = 0$), where the distortion is low but the rate is high, to auto-decoding behavior ($\beta = \infty$), where the distortion is high but the rate is low, all without having to change the model architecture. However, unlike Higgins et al. (2017); Alemi et al. (2017), we additionally optimize over the marginal $m(z)$ and compare across a variety of architectures, thus exploring a much larger solution space, which we illustrate empirically in Section 4.

## 3   RELATED WORK

Here we present an overview of the most closely related work. A more detailed treatment can be found in Appendix D.

**Model families for unsupervised learning with neural networks.**   There are two broad areas of active research in deep latent-variable models with neural networks: methods based on the variational autoencoder (VAE), introduced by Kingma & Welling (2014); Rezende et al. (2014), and

methods based on generative adversarial networks (GANs), introduced by Goodfellow et al. (2014). In this paper, we focus on the VAE family of models. In particular, we consider recent variants using inverse autoregressive flow (IAF) (Kingma et al., 2016), masked autoregressive flow (MAF) (Papamakarios et al., 2017), PixelCNN++ (Salimans et al., 2017), and the VampPrior (Tomczak & Welling, 2017), as well as common Conv/Deconv encoders and decoders.

**Information Theory and machine learning.** Barber & Agakov (2003) was the first to introduce tractable variational bounds on mutual information, and made close analogies and comparisons to maximum likelihood learning and variational autoencoders. The information bottleneck framework (Tishby et al., 1999; Shamir et al., 2010; Tishby & Zaslavsky, 2015; Alemi et al., 2017; Achille & Soatto, 2016; 2017) allows a model to smoothly trade off the minimality of the learned representation ($Z$) from data ($X$) by minimizing their mutual information, $I(X; Z)$, against the informativeness of the representation for the task at hand ($Y$) by maximizing their mutual information, $I(Z; Y)$. This constrained optimization problem is rephrased with the Lagrange multiplier, $\beta$, to the unconstrained optimization of $I(X; Z) - \beta I(Z; Y)$. Tishby & Zaslavsky (2015) plot an RD curve similar to the one in this paper, but they only consider the supervised setting, and they do not consider the information content that is implicit in powerful stochastic decoders. Higgins et al. (2017) proposed the $\beta$-VAE for unsupervised learning, which is a generalization of the original VAE in which the KL term is scaled by $\beta$, similar to this paper. However, they only considered $\beta > 1$. In this paper, we show that when using powerful autoregressive decoders, using $\beta \geq 1$ results in the model ignoring the latent code, so it is necessary to use $\beta < 1$.

**Generative Models and Compression.** Much recent work has explored the use of latent-variable generative models for image compression. Ballé et al. (2017) studies the problem explicitly in terms of the rate/distortion plane, adjusting a Lagrange multiplier on the distortion term to explore the convex hull of a model's optimal performance. Johnston et al. (2017) uses a recurrent VAE architecture to achieve state-of-the-art image compression rates, posing the loss as minimizing distortion at a fixed rate. Theis et al. (2017) writes the VAE loss as $R + \beta D$. Rippel & Bourdev (2017) shows that a GAN optimization procedure can also be applied to the problem of compression. All of these efforts focus on rate/distortion tradeoffs for individual models, but don't explore how the selection of the model itself affects the rate/distortion curve. Because we explore many combinations of modeling choices, we are able to more deeply understand how model selection impacts the rate/distortion curve, and to point out the area where all current models are lacking – the auto-encoding limit. Generative compression models also have to work with both quantized latent spaces and approximately fixed decoder model families trained with perceptual losses such as MS-SSIM (Wang et al., 2003), which constrain the form of the learned distribution. Our work does not assume either of these constraints are present for the tasks of interest.

## 4 EXPERIMENTS

**Toy Model**  In this section, we empirically show a case where the usual ELBO objective can learn a model which perfectly captures the true data distribution, $p^*(x)$, but which fails to learn a useful latent representation. However, by training the *same model* such that we minimize the distortion, subject to achieving a desired target rate $R^*$, we can recover a latent representation that closely matches the true generative process (up to a reparameterization), while also perfectly capturing the true data distribution.

We create a simple data generating process that consists of a true latent variable $Z^* = \{z_0, z_1\} \sim$ Ber(0.7) with added Gaussian noise and discretization. The magnitude of the noise was chosen so that the true generative model had I($x; z^*$) = 0.5 nats of mutual information between the observations and the latent. We additionally choose a model family with sufficient power to perfectly autoencode or autodecode. See Appendix E for more detail on the data generation and model.

Figure 2 shows various distributions computed using three models. For the left column, we use a hand-engineered encoder $e(z|x)$, decoder $d(x|z)$, and marginal $m(z)$ constructed with knowledge of the true data generating mechanism to illustrate an optimal model. For the middle and right columns, we learn $e(z|x)$, $d(x|z)$, and $m(z)$ using effectively infinite data sampled from $p^*(x)$ directly. The middle column is trained with ELBO. The right column is trained by targeting $R = 0.5$ while

minimizing $D$.[2] In both cases, we see that $p^*(x) \approx g(x) \approx d(x)$ for both trained models, indicating that optimization found the global optimum of the respective objectives. However, the VAE fails to learn a useful representation, only yielding a rate of $R = 0.0002$ nats,[3] while the Target Rate model achieves $R = 0.4999$ nats. Additionally, it nearly perfectly reproduces the true generative process, as can be seen by comparing the yellow and purple regions in the z-space plots (middle row) – both the optimal model and the Target Rate model have two clusters, one with about 70% of the probability mass, corresponding to class 0 (purple shaded region), and the other with about 30% of the mass (yellow shaded region) corresponding to class 1. In contrast, the z-space of the VAE completely mixes the yellow and purple regions, only learning a single cluster. Note that we reproduced essentially identical results with dozens of different random initializations for both the VAE and the Target Rate model – these results are not cherry-picked.

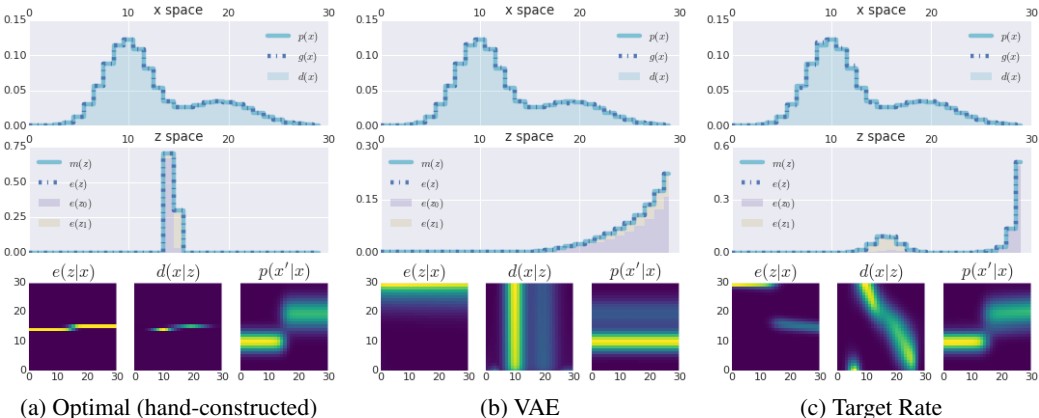

(a) Optimal (hand-constructed)    (b) VAE    (c) Target Rate

Figure 2: Toy Model illustrating the difference between fitting a model by maximizing ELBO (middle column) vs minimizing distortion for a fixed rate (right column). **Top:** Three distributions in data space: the true data distribution, $p^*(x)$, the model's generative distribution, $g(x) = \sum_z m(z)d(x|z)$, and the empirical data reconstruction distribution, $d(x) = \sum_{x'} \sum_z \hat{p}(x')e(z|x')d(x|z)$. **Middle:** Four distributions in latent space: the learned (or computed) marginal $m(z)$, the empirical induced marginal $e(z) = \sum_x \hat{p}(x)e(z|x)$, the empirical distribution over z values for data vectors in the set $\mathcal{X}_0 = \{x_n : z_n = 0\}$, which we denote by $e(z_0)$ in purple, and the empirical distribution over z values for data vectors in the set $\mathcal{X}_1 = \{x_n : z_n = 1\}$, which we denote by $e(z_1)$ in yellow. **Bottom:** Three $K \times K$ distributions: $e(z|x)$, $d(x|z)$ and $p(x'|x) = \sum_z e(z|x)d(x'|z)$.

**MNIST.**  In this section, we show how comparing models in terms of rate and distortion separately is more useful than simply observing marginal log likelihoods. We examine several VAE model architectures that have been proposed in the literature. We use the static binary MNIST dataset originally produced for (Larochelle & Murray, 2011)[4]. In appendix A, we show analogous results for the Omniglot dataset (Lake et al., 2015).

We will consider simple and complex variants for the encoder and decoder, and three different types of marginal. The simple encoder is a CNN with a fully factored 64 dimensional Gaussian for $e(z|x)$; the more complex encoder is similar, but followed by 4 steps of mean-only Gaussian inverse autoregressive flow (Kingma et al., 2016), with each step implemented as a 3 hidden layer MADE (Germain et al., 2015) with 640 units in each hidden layer. The simple decoder is a multi-layer deconvolutional network; the more powerful decoder is a PixelCNN++ (Salimans et al., 2017) model. The simple marginal is a fixed isotropic Gaussian, as is commonly used with VAEs; the more complicated version has a 4 step 3 layer MADE (Germain et al., 2015) mean-only Gaussian autoregressive flow (Papamakarios et al., 2017). We also consider the setting in which the marginal uses the VampPrior from (Tomczak & Welling, 2017). We will denote the particular model combination

---

[2] Note that the target value $R = \mathrm{I}(x; z^*) = 0.5$ is computed with knowledge of the true data generating distribution. However, this is the only information that is "leaked" to our method, and in general it is not hard to guess reasonable targets for $R$ for a given task and dataset.

[3] This is an example of VAEs ignoring the latent space. As decoder power increases, even $\beta = 1$ is sufficient to cause the model to collapse to the autoencoding limit.

[4] https://github.com/yburda/iwae/tree/master/datasets/BinaryMNIST

by the tuple $(+/-, +/-, +/ - /v)$, depending on whether we use a simple $(-)$ or complex $(+)$ (or $(v)$ VampPrior) version for the (encoder, decoder, marginal) respectively. In total we consider $2 \times 2 \times 3 = 12$ models. We train them all to minimize the objective in Equation 4. Full details can be found in Appendix F. Runs were performed at various values of $\beta$ ranging from 0.1 to 10.0, both with and without KL annealing (Bowman et al., 2016).

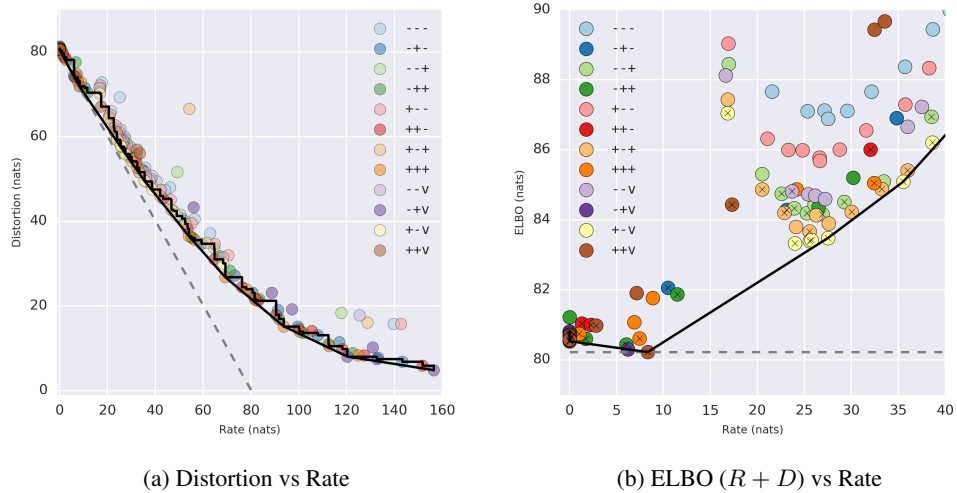

(a) Distortion vs Rate          (b) ELBO $(R + D)$ vs Rate

Figure 3: Results on MNIST. (a) The best achieved rate distortion value for each run plotted on the $RD$-plane. We denote the particular model combination by the tuple $(+/-, +/-, +/ - /v)$, depending on whether we use a simple $(-)$ or complex $(+)$ (or $(v)$ VampPrior) version for the (encoder, decoder, marginal) respectively. (b) The same data, but on the skew axes of ELBO = $R + D$ versus $R$.

**RD curve.** Figure 3a show the $RD$ plot for 12 models on the MNIST dataset. Dashed lines represent the best achieved test ELBO of 80.2 nats, which then sets an upper bound on the true data entropy $H$ for the static MNIST dataset. This implies that any $RD$ value above the dashed line is in principle achievable in a powerful enough model. The stepwise black curves show the monotonic Pareto frontier of achieved $RD$ points across all model families. Points participating in this curve are denoted with a $\times$ on the right. The grey solid line shows the corresponding convex hull, which we approach closely across all rates. Strong decoder model families dominate at the lowest and highest rates. Weak decoder models dominate at intermediate rates. Strong marginal models dominate strong encoder models at most rates. Across our model families we appear to be pushing up against an approximately smooth $RD$ curve. The 12 model families we considered here, arguably a representation of the classes of models considered in the VAE literature, in general perform much worse in the auto-encoding limit (bottom right corner) of the $RD$ plane. This is likely due to a lack of power in our current marginal approximations.

Figure 3b shows the same raw data, but where we plot ELBO=$R + D$ versus $R$. Here some of the differences between individual model families performances are more easily resolved. Broadly, models with a deconvolutional decoder perform well at intermediate ~22 nat rates, but quickly suffer large distortion penalties as they move away from that point. This is perhaps unsurprising considering we trained on the binary MNIST dataset, for which the measured pixel level sampling entropy on the test set is approximately 22 nats.

Models with a powerful autoregressive decoder perform well at low rates, but for values of $\beta \geq 1$ tend to collapse to pure autodecoding models. With the use of the VampPrior and KL annealing however, $\beta = 1$ models can exist at finite rates of around 8 nats. Our framework helps explain the observed difficulties in the literature of training a useful VAE with a powerful decoder, and the observed utility of techniques like "free bits" (Kingma et al., 2016), "soft free bits" (Chen et al., 2017) and KL annealing (Bowman et al., 2016). Each of these effectively trains at a reduced $\beta$, moving up along the $RD$ curve. Without any additional modifications, simply training at reduced

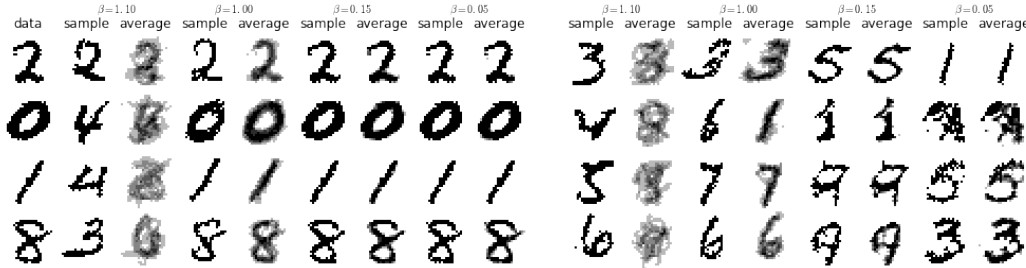

(a) MNIST Reconstructions: $z \sim e(z|x), \hat{x} \sim d(x|z)$     (b) MNIST Generations: $z \sim m(z), \hat{x} \sim d(x|z)$

Figure 4: We can smoothly move between pure autodecoding and autoencoding behavior in a single model family by tuning $\beta$. (a) Sampled reconstructions from the -+v model family trained at given $\beta$ values. Pairs of columns show a single reconstruction and the mean of 5 reconstructions. The first column shows the input samples. (b) Generated images from the same set of models. The pairs of columns are single samples and the mean of 5 samples. See text for discussion.

$\beta$ is a simpler way to achieve nonvanishing rates, without additional architectual adjustments like in the variational lossy autoencoder (Chen et al., 2017).

Analyzing model performance using the $RD$ curve gives a much more insightful comparison of relative model performance than simply comparing marginal data log likelihoods. In particular, we managed to achieve models with five-sample IWAE (Burda et al., 2015) estimates below 82 nats (a competitive rate for single layer latent variable models (Tomczak & Welling, 2017)) for rates spanning from $10^{-4}$ to 30 nats. While all of those models have competitive ELBOs or marginal log likelihood, they differ substantially in the tradeoffs they make between rate and distortion, and those differences result in qualitatively different model behavior, as illustrated in Figure 4.

**The interaction between latent variables and powerful decoders.** Within any particular model family, we can smoothly move between and explore its performance at varying rates. An illustrative example is shown in Fig. 4, where we study the effect of changing $\beta$ (using KL annealing from low to high) on the same -+v model, corresponding to a VAE with a simple encoder, a powerful PixelCNN++ decoder, and a powerful VampPrior marginal.

In Fig. 4a we assess how well the models do at reconstructing their inputs. We pick an image $x$ at random, encode it using $z \sim e(z|x)$, and then reconstruct it using $\hat{x} \sim d(x|z)$. When $\beta = 1.10$ (left column), the model obtains $R = 0.0004, D = 80.6, \text{ELBO} = 80.6$ nats. The tiny rate indicates that the decoder ignores its latent code, and hence the reconstructions are independent of the input $x$. For example, when the input is $x = 8$, the reconstruction is $\hat{x} = 3$. However, the generated images sampled from the decoder look good (this is an example of an autodecoder). At the other extreme, when $\beta = 0.05$ (right column), the model obtains $R = 156, D = 4.8$, ELBO=161 nats. Here the model does an excellent job of auto-encoding, generating nearly pixel perfect reconstructions. However, samples from this model's prior, as shown on the right, are of very poor quality, reflected in the worse ELBO and IWAE values. At intermediate values, such as $\beta = 1.0$, ($R = 6.2, D = 74.1$, ELBO=80.3) the model seems to retain semantically meaningful information about the input, such as its class and width of the strokes, but maintains variation in the individual reconstructions. In particular, notice that the individual "2" sent in is reconstructed as a similar "2" but with a visible loop at the bottom. This model also has very good generated samples. This intermediate rate encoding arguably typifies what we want to achieve in unsupervised learning: we have learned a highly compressed representation that retains salient features of the data. In the third column, the $\beta = 0.15$ model ($R = 120.3, D = 8.1$, ELBO=128) we have very good reconstructions Figure 4b one can visually inspect while still obtaining a good degree of compression. This model arguably typifies the domain most compression work is interested in, where most perceivable variations in the digit are retained in the compression. However, at these higher rates the failures of our current architectures to approach their theoretical performance becomes more apparent, as the corresponding ELBO of 128 nats is much higher than the 81 nats we obtain at low rates. This is also evident in the visual degradation in the generated samples.

While it is popular to visualize both the reconstructions and generated samples from VAEs, we suggest researchers visually compare several sampled decodings using the same sample of the latent variable, whether it be from the encoder or the prior, as done here in Figure 4. By using a single sample of the latent variable, but decoding it multiple times, one can visually inspect what features of the input are captured in the observed value for the rate. This is particularly important to do when using powerful decoders, such as autoregressive models.

## 5 DISCUSSION AND FURTHER WORK

We have motivated the $\beta$-VAE objective on information theoretic grounds, and demonstrated that comparing model architectures in terms of the rate-distortion plot offers a much better look at their performance and tradeoffs than simply comparing their marginal log likelihoods. Additionally, we have shown a simple way to fix models that ignore the latent space due to the use of a powerful decoder: simply reduce $\beta$ and retrain. This fix is much easier to implement than other solutions that have been proposed in the literature, and comes with a clear theoretical justification. We strongly encourage future work to report rate and distortion values independently, rather than just reporting the log likelihood. If future work proposes new architectural regularization techniques, we suggest the authors train their objective at various rate distortion tradeoffs to demonstrate and quantify the region of the RD plane where their method dominates.

Through a large set of experiments we have demonstrated the performance at various rates and distortion tradeoffs for a set of representative architectures currently under study, confirming the power of autoregressive decoders, especially at low rates. We have also shown that current approaches seem to have a hard time achieving high rates at low distortion. This suggests a set of experiments with a simple encoder / decoder pair but a powerful autoregressive marginal posterior approximation, which should in principle be able to reach the autoencoding limit, with vanishing distortion and rates approaching the data entropy.

Interpreting the $\beta$-VAE objective as a constrained optimization problem also hints at the possibility of applying more powerful constrained optimization techniques, which we hope will be able to advance the state of the art in unsupervised representation learning.

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

## A  RESULTS ON OMNIGLOT

Figure 5 plots the RD curve for various models fit to the Omniglot dataset (Lake et al., 2015), in the same form as the MNIST results in Figure 3. Here we explored $\beta$s for the powerful decoder models ranging from 1.1 to 0.1, and $\beta$s of 0.9, 1.0, and 1.1 for the weaker decoder models.

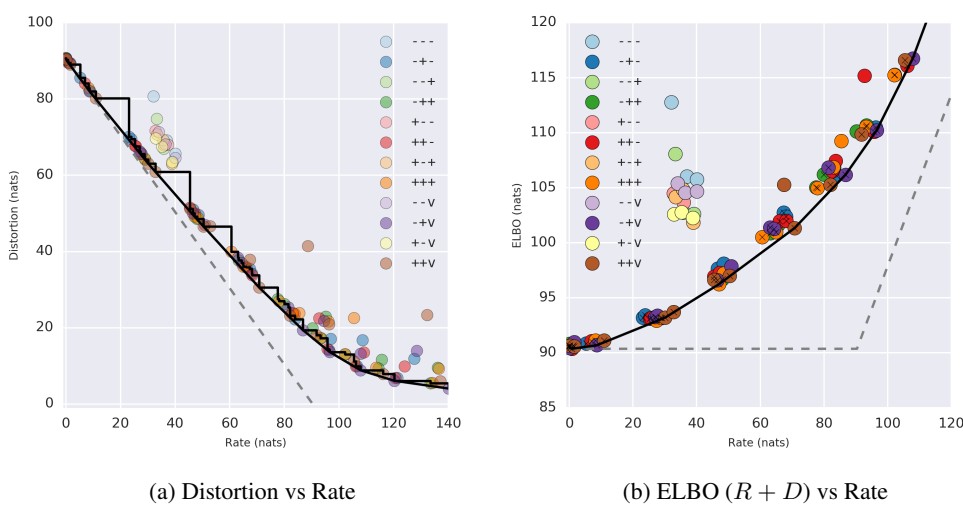

(a) Distortion vs Rate                    (b) ELBO ($R + D$) vs Rate

Figure 5: Results on Omniglot. Otherwise same description as Figure 3. (a) Rate-distortion curves. (b) The same data, but on the skew axes of ELBO = $R + D$ versus $R$.

On Omniglot, the powerful decoder models dominate over the weaker decoder models. The powerful decoder models with their autoregressive form most naturally sit at very low rates. We were able to obtain finite rates by means of KL annealing. Further experiments will help to fill in the details especially as we explore differing $\beta$ values for these architectures on the Omniglot dataset. Our best achieved ELBO was at 90.37 nats, set by the ++- model with $\beta = 1.0$ and KL annealing. This model obtains $R = 0.77, D = 89.60, ELBO = 90.37$ and is nearly auto-decoding. We found 14 models with ELBOs below 91.2 nats ranging in rates from 0.0074 nats to 10.92 nats.

Similar to Figure 4 in Figure 6 we show sample reconstruction and generated images from the same "-+v" model family trained with KL annealing but at various $\beta$s. Just like in the MNIST case, this demonstrates that we can smoothly interpolate between auto-decoding and auto-encoding behavior in a single model family, simply by adjusting the $\beta$ value.

## B  PROOFS

### B.1  LOWER BOUND ON REPRESENTATIONAL MUTUAL INFORMATION

Our lower bound is established by the fact that Kullback-Leibler (KL) divergences are positive semidefinite

$$\text{KL}[q(x|z) \,||\, p(x|z)] = \int dx \, q(x|z) \log \frac{q(x|z)}{p(x|z)} \geq 0$$

which implies for any distribution $p(x|z)$:

$$\int dx \, q(x|z) \log q(x|z) \geq \int dx \, q(x|z) \log p(x|z)$$

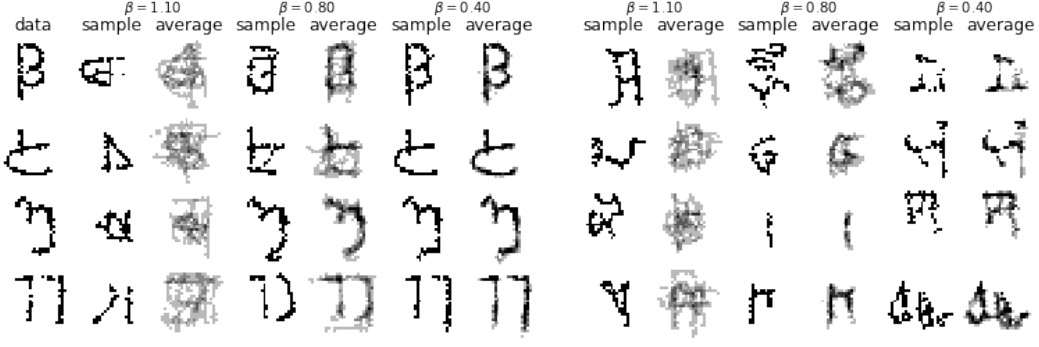

(a) Omniglot Reconstructions: $z \sim e(z|x)$, $\hat{x} \sim d(x|z)$      (b) Omniglot Generations: $z \sim m(z)$, $\hat{x} \sim d(x|z)$

Figure 6: We can smoothly move between pure autodecoding and autoencoding behavior in a single model family by tuning $\beta$. (a) Sampled reconstructions from the -+v model family trained at given $\beta$ values. Pairs of columns show a single reconstruction and the mean of 5 reconstructions. The first column shows the input samples. (b) Generated images from the same set of models. The pairs of columns are single samples and the mean of 5 samples. See text for discussion.

$$
\begin{aligned}
I_{\text{rep}} = I_{\text{rep}}(X;Z) &= \iint dx\, dz\, p_e(x,z) \log \frac{p_e(x,z)}{p^*(x)p_e(z)} \\
&= \int dz\, p_e(z) \int dx\, p_e(x|z) \log \frac{p_e(x|z)}{p^*(x)} \\
&= \int dz\, p_e(z) \left[ \int dx\, p_e(x|z) \log p_e(x|z) - \int dx\, p_e(x|z) \log p^*(x) \right] \\
&\geq \int dz\, p_e(z) \left[ \int dx\, p_e(x|z) \log d(x|z) - \int dx\, p_e(x|z) \log p^*(x) \right] \\
&= \iint dx\, dz\, p_e(x,z) \log \frac{d(x|z)}{p^*(x)} \\
&= \int dx\, p^*(x) \int dz\, e(z|x) \log \frac{d(x|z)}{p^*(x)} \\
&= \left( -\int dx\, p^*(x) \log p^*(x) \right) - \left( -\int dx\, p^*(x) \int dz\, e(z|x) \log d(x|z) \right) \\
&\equiv H - D
\end{aligned}
$$

## B.2   Upper Bound on Representational Mutual Information

The upper bound is established again by the positive semidefinite quality of KL divergence.

$$
\text{KL}[q(z|x) \,||\, p(z)] \geq 0 \implies \int dz\, q(z|x) \log q(z|x) \geq \int dz\, q(z|x) \log p(z)
$$

$$\begin{aligned}
\mathrm{I}_{\mathrm{rep}} = \mathrm{I}_{\mathrm{rep}}(X; Z) &= \iint dx\, dz\, p_e(x, z) \log \frac{p_e(x, z)}{p^*(x) p_e(z)} \\
&= \iint dx\, dz\, p_e(x, z) \log \frac{e(z|x)}{p_e(z)} \\
&= \iint dx\, dz\, p_e(x, z) \log e(z|x) - \iint dx\, dz\, p_e(x, z) \log p_e(z) \\
&= \iint dx\, dz\, p_e(x, z) \log e(z|x) - \int dz\, p_e(z) \log p_e(z) \\
&\leq \iint dx\, dz\, p_e(x, z) \log e(z|x) - \int dz\, p_e(z) \log m(z) \\
&= \iint dx\, dz\, p_e(x, z) \log e(z|x) - \iint dx\, dz\, p_e(x, z) \log m(z) \\
&= \iint dx\, dz\, p_e(x, z) \log \frac{e(z|x)}{m(z)} \\
&= \int dx\, p^*(x) \int dz\, e(z|x) \log \frac{e(z|x)}{m(z)} \equiv R
\end{aligned}$$

## B.3 OPTIMAL MARGINAL FOR FIXED ENCODER

Here we establish that the optimal marginal approximation $p(z)$, is precisely the marginal distribution of the encoder.

$$R \equiv \int dx\, p^*(x) \int dz\, e(z|x) \log \frac{e(z|x)}{m(z)}$$

Consider the variational derivative of the rate with respect to the marginal approximation:

$$m(z) \to m(z) + \delta m(z) \quad \int dz\, \delta m(z) = 0$$

$$\begin{aligned}
\delta R &= \int dx\, p^*(x) \int dz\, e(z|x) \log \frac{e(z|x)}{m(z) + \delta m(z)} - R \\
&= \int dx\, p^*(x) \int dz\, e(z|x) \log \left( 1 + \frac{\delta m(z)}{m(z)} \right) \\
&\sim \int dx\, p^*(x) \int dz\, e(z|x) \frac{\delta m(z)}{m(z)}
\end{aligned}$$

Where in the last line we have taken the first order variation, which must vanish if the total variation is to vanish. In particular, in order for this variation to vanish, since we are considering an arbitrary $\delta m(z)$, except for the fact that the integral of this variation must vanish, in order for the first order variation in the rate to vanish it must be true that for every value of $x, z$ we have that:

$$m(z) \propto p^*(x) e(z|x),$$

which when normalized gives:

$$m(z) = \int dx\, p^*(x) e(z|x),$$

or that the marginal approximation is the true encoder marginal.

## B.4 OPTIMAL DECODER FOR FIXED ENCODER

Next consider the variation in the distortion in terms of the decoding distribution with a fixed encoding distribution.

$$d(x|z) \to d(x|z) + \delta d(x|z) \quad \int dx\, d(x|z) = 0$$

$$\delta D = -\int dx\, p^*(x) \int dz\, e(z|x) \log(d(x|z) + \delta d(x|z)) - D$$

$$= -\int dx\, p^*(x) \int dz\, e(z|x) \log\left(1 + \frac{\delta d(x|z)}{d(x|z)}\right)$$

$$\sim -\int dx\, p^*(x) \int dz\, e(z|x) \frac{\delta d(x|z)}{d(x|z)}$$

Similar to the section above, we took only the leading variation into account, which itself must vanish for the full variation to vanish. Since our variation in the decoder must integrate to 0, this term will vanish for every $x, z$ we have that:

$$d(x|z) \propto p^*(x)e(z|x),$$

when normalized this gives:

$$d(x|z) = e(z|x) \frac{p^*(x)}{\int dx\, p^*(x)e(z|x)}$$

which ensures that our decoding distribution is the correct posterior induced by our data and encoder.

## B.5 LOWER BOUND ON GENERATIVE MUTUAL INFORMATION

The lower bound is established as all other bounds have been established, with the positive semidefiniteness of KL divergences.

$$\mathrm{KL}[d(z|x)\,||\,q(z|x)] = \int dz\, d(z|x) \log \frac{d(z|x)}{q(z|x)} \geq 0$$

which implies for any distribution $q(z|x)$:

$$\int dz\, d(z|x) \log d(z|x) \geq \int dz\, d(z|x) \log q(z|x)$$

$$I_{\mathrm{gen}} = I_{\mathrm{gen}}(X; Z) = \iint dx\, dz\, p_{\mathrm{gen}}(x, z) \log \frac{p_{\mathrm{gen}}(x, z)}{p_{\mathrm{gen}}(x)p_{\mathrm{gen}}(z)}$$

$$= \int dx\, p_{\mathrm{gen}}(x) \int dz\, p_{\mathrm{gen}}(z|x) \log \frac{p_{\mathrm{gen}}(z|x)}{m(z)}$$

$$= \int dx\, p_{\mathrm{gen}}(x) \left[\int dz\, p_{\mathrm{gen}}(z|x) \log p_{\mathrm{gen}}(z|x) - \int dz\, p_{\mathrm{gen}}(z|x) \log m(z)\right]$$

$$\geq \int dx\, p_{\mathrm{gen}}(x) \left[\int dz\, p_{\mathrm{gen}}(z|x) \log e(z|x) - \int dz\, p_{\mathrm{gen}}(z|x) \log m(z)\right]$$

$$= \iint dx\, dz\, p_{\mathrm{gen}}(x, z) \log \frac{e(z|x)}{m(z)}$$

$$= \int dz\, m(z) \int dx\, d(x|z) \log \frac{e(z|x)}{m(z)}$$

$$\equiv E$$

## B.6 UPPER BOUND ON GENERATIVE MUTUAL INFORMATION

The upper bound is establish again by the positive semidefinite quality of KL divergence.

$$\mathrm{KL}[p(x|z)\,||\,r(x)] \geq 0 \implies \int dx\, p(x|z) \log p(x|z) \geq \int dx\, p(x|z) \log r(x)$$

$$
\begin{aligned}
I_{\text{gen}} = I_{\text{gen}}(X; Z) &= \iint dx\, dz\, p_{\text{gen}}(x, z) \log \frac{p_{\text{gen}}(x, z)}{p_{\text{gen}}(x) m(z)} \\
&= \iint dx\, dz\, p_{\text{gen}}(x, z) \log \frac{d(x|z)}{p_{\text{gen}}(x)} \\
&= \iint dx\, dz\, p_{\text{gen}}(x, z) \log d(x|z) - \iint dx\, dz\, p_{\text{gen}}(x, z) \log p_{\text{gen}}(x) \\
&= \iint dx\, dz\, p_{\text{gen}}(x, z) \log d(x|z) - \int dx\, p_{\text{gen}}(x) \log p_{\text{gen}}(x) \\
&\leq \iint dx\, dz\, p_{\text{gen}}(x, z) \log d(x|z) - \int dx\, p_{\text{gen}}(x) \log q(x) \\
&= \iint dx\, dz\, p_{\text{gen}}(x, z) \log d(x|z) - \iint dx\, dz\, p_{\text{gen}}(x, z) \log q(x) \\
&= \iint dx\, dz\, p_{\text{gen}}(x, z) \log \frac{d(x|z)}{q(x)} \\
&= \int dz\, m(z) \int dx\, d(x|z) \log \frac{d(x|z)}{q(x)} \equiv G
\end{aligned}
$$

## C    GENERATIVE MUTUAL INFORMATION

Given any four distributions: $p^*(x)$ – a density over some data space $X$, $e(z|x)$ – a stochastic map from that data to a new representational space $Z$, $d(x|z)$ – a stochastic map in the reverse direction from $Z$ to $X$, and $m(z)$ – some density in the $Z$ space; we were able to find an inequality relating three functionals of these densities that must always hold. We found this inequality by deriving upper and lower bounds on the mutual information in the joint density defined by the natural *representational* path through the four distributions, $p_e(x, z) = p^*(x) e(z|x)$. Doing so naturally made us consider the existence of two other distributions $d(x|z)$ and $m(z)$. Let's consider the mutual information along this new *generative* path.

$$
p_{\text{gen}}(x, z) = m(z) d(x|z) \tag{5}
$$

$$
I_{\text{gen}}(X; Z) = \iint dx\, dz\, p_{\text{gen}}(x, z) \log \frac{p_{\text{gen}}(x, z)}{p_{\text{gen}}(x) p_{\text{gen}}(z)} \tag{6}
$$

Just as before we can easily establish both a variational lower and upper bound on this mutual information. For the lower bound (proved in Section B.5), we have:

$$
E \equiv \int dz\, p(z) \int dx\, p(x|z) \log \frac{q(z|x)}{p(z)} \leq I_{\text{gen}} \tag{7}
$$

Where we need to make a variational approximation to the decoder posterior, itself a distribution mapping $X$ to $Z$. Since we already have such a distribution from our other considerations, we can certainly use the encoding distribution $q(z|x)$ for this purpose, and since the bound holds for any choice it will hold with this choice. We will call this bound $E$ since it gives the distortion as measured through the *encoder* as it attempts to encode the generated samples back to their latent representation.

We can also find a variational upper bound on the generative mutual information (proved in Section B.6):

$$
G \equiv \int dz\, m(z) \int dx\, d(x|z) \log \frac{d(x|z)}{q(x)} \geq I_{\text{gen}} \tag{8}
$$

This time we need a variational approximation to the marginal density of our generative model, which we denote as $q(x)$. We call this bound $G$ for the rate in the *generative* model.

Together these establish both lower and upper bounds on the generative mutual information:

$$
E \leq I_{\text{gen}} \leq G. \tag{9}
$$

In our early experiments, it appears as though additionally constraining or targeting values for these generative mutual information bounds is important to ensure consistency in the underlying joint distributions. In particular, we notice a tendency of models trained with the $\beta$-VAE objective to have loose bounds on the generative mutual information when $\beta$ varies away from 1.

## C.1 Rearranging the Representational Lower Bound

In light of the appearance of a new independent density estimate $q(x)$ in deriving our variational upper bound on the mutual information in the generative model, let's actually use that to rearrange our variational lower bound on the representational mutual information.

$$\int dx\, p^*(x) \int dz\, e(z|x) \log \frac{e(z|x)}{p^*(x)} = \int dx\, p^*(x) \int dz\, e(z|x) \log \frac{e(z|x)}{q(x)} - \int dx\, p^*(x) \log \frac{p^*(x)}{q(x)} \tag{10}$$

Doing this, we can express our lower bound in terms of two reparameterization independent functionals:

$$U \equiv \int dx\, p^*(x) \int dz\, e(z|x) \log \frac{d(x|z)}{q(x)} \tag{11}$$

$$S \equiv \int dx\, p^*(x) \log \frac{p^*(x)}{q(x)} = -\int dx\, p^*(x) \log q(x) - H \tag{12}$$

This new reparameterization couples together the bounds we derived both the representational mutual information and the generative mutual information, using $q(x)$ in both. The new function $S$ we've described is intractable on its own, but when split into the data entropy and a cross entropy term, suggests we set a target cross entropy on our own density estimate $q(x)$ with respect to the empirical data distribution that might be finite in the case of finite data.

Together we have an equivalent way to formulate our original bounds on the representaional mutual information

$$U - S = H - D \leq I_{\text{rep}} \leq R \tag{13}$$

We believe this reparameterization offers and important and potential way to directly control for overfitting. In particular, given that we compute our objectives using a finite sample from the true data distribution, it will generically be true that $\text{KL}[\hat{p}(x) \,||\, p^*(x)] \geq 0$. In particular, the usual mode we operate in is one in which we only ever observe each example once in the training set, suggesting that in particular an estimate for this divergence would be:

$$\text{KL}[\hat{p}(x) \,||\, p^*(x)] \sim H(X) - \log N. \tag{14}$$

Early experiments suggest this offers a useful target for $S$ in the reparameterized objective that can prevent overfitting, at least in our toy problems.

## D Detailed Related Work

Here we expand on the brief related work in Section 3.

### D.1 Information Theory and machine learning

Much recent work has leveraged information theory to improve our understanding of machine learning in general, and unsupervised learning specifically. In Tishby & Zaslavsky (2015), the authors present theory for the success of supervised deep learning as approximately optimizing the information bottleneck objective, and also theoretically predict a supervised variant of the rate/distortion plane we describe here. Shwartz-Ziv & Tishby (2017) further proposes that training of deep learning models proceeds in two phases: error minimization followed by compression. They suggest that the compression phase diffuses the conditional entropy of the individual layers of the model, and when the model has converged, it lies near the information bottleneck optimal frontier on the proposed rate/distortion plane. In Higgins et al. (2016) the authors motivate the $\beta$-VAE objective from a combined neuroscience and information theoretic perspective. The Higgins et al. (2017) propose that $\beta$ should be greater than 1 to properly learn disentangled representations in an unsupervised manner.

Chen et al. (2017) described the issue of the too-powerful decoder when training standard VAEs, where $\beta = 1$. They proposed a bits-back (Hinton & Van Camp, 1993) model to understand this phenomenon, as well as a noise-injection technique to combat it. Our approach removes the need for an additional noise source in the decoder, and concisely rephrases the problem as finding the optimal $\beta$ for the chosen model family, which can now be as powerful as we like, without risk of ignoring the latent space and collapsing to an autodecoding model.

Bowman et al. (2016) suggested annealing the weight of the KL term of the ELBO ($\mathrm{KL}[q(z|x) \parallel p(z)]$) from 0 to 1 to make it possible to train an RNN decoder without ignoring the latent space. Snderby et al. (2016) applies the same idea to ladder network decoders. We relate this idea of KL annealing to our optimal rate/distortion curve, and show empirically that KL annealing does not in general attain the performance possible when setting a fixed $\beta$ or a fixed target rate.

In Achille & Soatto (2016), the authors proposed an information bottleneck approach to the activations of a network, termed Information Dropout, as a form of regularization that explains and generalizes Dropout (Srivastava et al., 2014). They suggest that, without such a form of regularization, standard SGD training only provides a sufficient statistic, but does not in general provide two other desiderata: minimality and invariance to nuisance factors. Both of these would be provided by a procedure that directly optimized the information bottleneck. They propose that simply injecting noise adaptively into the otherwise deterministic function computed by the deep network is sufficient to cause SGD to optimize toward disentangled and invariant representations. Achille & Soatto (2017) expands on this exploration of sufficiency, minimality, and invariance in deep networks. In particular they propose that architectural bottlenecks and depth both promote invariance directly, and they decompose the standard cross entropy loss used in supervised learning into four terms, including one which they name 'overfitting', and which, without other regularization, an optimization procedure can easily increase in order to reduce the total loss.

Other recent work explores related theoretical frameworks for unsupervised learning, including Pu et al. (2017); Hu et al. (2017); Zhang et al. (2017).

## D.2 MODEL FAMILIES FOR UNSUPERVISED LEARNING WITH NEURAL NETWORKS

Burda et al. (2015) presented an importance-weighted variant of the VAE objective. By increasing the number of samples taken from the encoder during training, they are able to tighten the variational lower bound and improve the test log likelihood.

Rezende & Mohamed (2015) proposed to use normalizing flows to approximate the true posterior during inference, in order to overcome the problem of the standard mean-field posterior approximation used in VAEs lacking sufficient representational power to model complex posterior distributions. Normalizing flow permits the use of a deep network to compute a differentiable function with a computable determinant of a random variable and have the resulting function be a valid normalized distribution. Kingma et al. (2016) expanded on this idea by introducing inverse autoregressive flow (IAF). IAF takes advantage of properties of current autoregressive models, including their expressive power and particulars of their Jacobians when inverted, and used them to learn expressive, parallelizeable normalizing flows that are efficient to compute when using high dimensional latent spaces for the posterior.

Autoregressive models have also been applied successfully to the density estimation problem, as well as high quality sample generation. MADE (Germain et al., 2015) proposed directly masking the parameters of an autoencoder during generation such that a given unit makes its predictions based solely on the first $d$ activations of the layer below. This enforces that the autoencoder maintains the "autoregressive" property. In Oord et al. (2016), the authors presented a recurrent neural network that can autoregressively predict the pixels of an image, as well as provide tractable density estimation. This work was expanded to a convolutional model called PixelCNN (van den Oord et al., 2016), which enforced the autoregressive property by masking the convolutional filters. In Salimans et al. (2017), the authors further improved the performance with PixelCNN++ with a collection of architecture changes that allow for much faster training. Finally, Papamakarios et al. (2017) proposed another unification of normalizing flow models with autoregressive models for density estimation. The authors observe that the conditional ordering constraints required for valid autoregressive modeling enforces a choice which may be arbitrarily incorrect for any particular problem. In their proposal, Masked Autoregressive Flow (MAF), they explicitly model the random number

generation process with stacked MADE layers. This particular choice means that MAF is fast at density estimation, whereas the nearly identical IAF architecture is fast at sampling.

Tomczak & Welling (2017) proposed a novel method for learning the marginal posterior, $m(z)$ (written $q(z)$ in that work): learn $k$ *pseudo-inputs* that can be mixed to approximate any of the true samples $x \sim p^*(x)$.

## E  TOY MODEL DETAILS

**Data generation.**   The true data generating distribution is as follows. We first sample a latent binary variable, $z \sim \text{Ber}(0.7)$, then sample a latent 1d continuous value from that variable, $h|z \sim \mathcal{N}(h|\mu_z, \sigma_z)$, and finally we observe a discretized value, $x = \text{discretize}(h; \mathcal{B})$, where $\mathcal{B}$ is a set of 30 equally spaced bins. We set $\mu_z$ and $\sigma_z$ such that $R^* \equiv \text{I}(x; z) = 0.5$ nats, in the true generative process, representing the ideal rate target for a latent variable model.

**Model details.**   We choose to use a discrete latent representation with $K = 30$ values, with an encoder of the form $e(z_i|x_j) \propto -\exp[(w_i^e x_j - b_i^e)^2]$, where $z$ is the one-hot encoding of the latent categorical variable, and $x$ is the one-hot encoding of the observed categorical variable. Thus the encoder has $2K = 60$ parameters. We use a decoder of the same form, but with different parameters: $d(x_j|z_i) \propto -\exp[(w_i^d x_j - b_i^d)^2]$. Finally, we use a variational marginal, $m(z_i) = \pi_i$. Given this, the true joint distribution has the form $p_e(x, z) = p^*(x)e(z|x)$, with marginal $m(z) = \sum_x p_e(x, z)$ and conditional $p_e(x|z) = p_e(x, z)/p_e(z)$.

## F  DETAILS FOR MNIST AND OMNIGLOT EXPERIMENTS

We used the static binary MNIST dataset originally produced for (Larochelle & Murray, 2011)[5], and the Omniglot dataset from Lake et al. (2015); Burda et al. (2015).

As stated in the main text, for our experiments we considered twelve different model families corresponding to a simple and complex choice for the encoder and decoder and three different choices for the marginal.

Unless otherwise specified, all layers used a linearly gated activation function activation function (Dauphin et al., 2017), $h(x) = (W_1 x + b_2)\sigma(W_2 x + b_2)$.

### F.1  ENCODER ARCHITECTURES

For the encoder, the simple encoder was a convolutional encoder outputting parameters to a diagonal Gaussian distribution. The inputs were first transformed to be between -1 and 1. The architecture contained 5 convolutional layers, summarized in the format Conv (depth, kernel size, stride, padding), followed by a linear layer to read out the mean and a linear layer with softplus nonlinearity to read out the variance of the diagonal Gaussiann distribution.

- Input (28, 28, 1)
- Conv (32, 5, 1, same)
- Conv (32, 5, 2, same)
- Conv (64, 5, 1, same)
- Conv (64, 5, 2, same)
- Conv (256, 7, 1, valid)
- Gauss (Linear (64), Softplus (Linear (64)))

For the more complicated encoder, the same 5 convolutional layer architecture was used, followed by 4 steps of mean-only Gaussian inverse autoregressive flow, with each step's location parameters computed using a 3 layer MADE style masked network with 640 units in the hidden layers and ReLU activations.

---

[5]https://github.com/yburda/iwae/tree/master/datasets/BinaryMNIST

## F.2 Decoder architectures

The simple decoder was a transposed convolutional network, with 6 layers of transposed convolution, denoted as Deconv (depth, kernel size, stride, padding) followed by a linear convolutional layer parameterizing an independent Bernoulli distribution over all of the pixels:

- Input (1, 1, 64)
- Deconv (64, 7, 1, valid)
- Deconv (64, 5, 1, same)
- Deconv (64, 5, 2, same)
- Deconv (32, 5, 1, same)
- Deconv (32, 5, 2, same)
- Deconv (32, 4, 1, same)
- Bernoulli (Linear Conv (1, 5, 1, same))

The complicated decoder was a slightly modified PixelCNN++ style network (Salimans et al., 2017)[6]. However in place of the original RELU activation functions we used linearly gated activation functions and used six blocks (with sizes $(28 \times 28) - (14 \times 14) - (7 \times 7) - (7 \times 7) - (14 \times 14) - (28 \times 28)$) of two resnet layers in each block. All internal layers had a feature depth of 64. Shortcut connections were used throughout between matching sized featured maps. The 64-dimensional latent representation was sent through a dense lineary gated layer to produce a 784-dimensional representation that was reshaped to $(28 \times 28 \times 1)$ and concatenated with the target image to produce a $(28 \times 28 \times 2)$ dimensional input. The final output (of size $(28 \times 28 \times 64)$) was sent through a $(1 \times 1)$ convolution down to depth 1. These were interpreted as the logits for a Bernoulli distribution defined on each pixel.

## F.3 Marginal architectures

We used three different types of marginals. The simplest architecture (denoted (-)), was just a fixed isotropic gaussian distribution in 64 dimensions with means fixed at 0 and variance fixed at 1.

The complicated marginal (+) was created by transforming the isotropic Gaussian base distribution with 4 layers of mean-only Gaussian autoregressive flow, with each steps location parameters computed using a 3 layer MADE style masked network with 640 units in the hidden layers and relu activations. This network resembles the architecture used in Papamakarios et al. (2017).

The last choice of marginal was based on VampPrior and denoted with (v), which uses a mixture of the encoder distributions computed on a set of pseudo-inputs to parameterize the prior (Tomczak & Welling, 2017). We add an additional learned set of weights on the mixture distributions that are constrained to sum to one using a softmax function: $m(z) = \sum_{i=1}^{N} w_i e(z|\phi_i)$ where $N$ are the number of pseudo-inputs, $w$ are the weights, $e$ is the encoder, and $\phi$ are the pseudo-inputs that have the same dimensionality as the inputs.

## F.4 Optimization

The models were all trained using the $\beta$-VAE objective (Higgins et al., 2017) at various values of $\beta$. No form of explicit regularization was used. The models were trained with Adam (Kingma & Ba, 2015) with normalized gradients (Yu et al., 2017) for 200 epochs to get good convergence on the training set, with a fixed learning rate of $3 \times 10^{-4}$ for the first 100 epochs and a linearly decreasing learning rate towards 0 at the 200th epoch.

---

[6]Original implmentation available at https://github.com/openai/pixel-cnn

