# OpenReview forum: "An information-theoretic analysis of deep latent-variable models"
_ICLR.cc/2018/Conference — Reject_

### Official Review · AnonReviewer2 · 2017-11-26
**Potentially interesting but insights not clear**

**Rating:** 5
**Confidence:** 4

**Review:**

Summary:

This paper optimizes the beta-VAE objective and analyzes the resulting models in terms of the two components of the VAE loss: the reconstruction error (which the authors refer to as distortion, “D”) and the KL divergence term (which the authors refer to as rate, “R”). Various VAEs using either PixelCNN++ or a simpler model for the encoder, decoder, or marginal distribution of a VAE are trained on MNIST (with some additional results on OMNIGLOT) and analyzed in terms of samples, reconstructions, and their rate-distortion trade-off.

Review:

I find it difficult to point my finger to novel conceptual or theoretical insights in this paper. The idea of maximizing information for unsupervised learning of representations has of course been explored a lot (e.g., Bell & Sejnowski, 1995). Deeper connections between variational inference and rate-distortion have been made before (e.g., Balle et al., 2017; Theis et al., 2017), while this paper merely seems to rename the reconstruction and KL terms of the ELBO. Variational lower and upper bounds on mutual information have been used before as well (e.g., Barber & Agakov, 2003; Alemi et al., 2017), although they are introduced like new results in this paper. The derived “sandwich equation” only seems to be used to show that H - D - R <= 0, which also follows directly from Gibbs’ inequality (since the left-hand side is a negative KL divergence). The main contributions therefore seem to be the proposed analysis of models in the R-D plane, and the empirical contribution of analyzing beta-VAEs.

Based on the R-D plots, the authors identify a potential problem of generative models, namely that none of the trained models appear to get close to the “auto-encoding limit” where the distortion is close zero. Wouldn’t this gap easily be closed by a model with identity encoder, identity decoder, and PixelCNN++ for the marginal distribution? Given that autoregressive models generally perform better than VAEs in terms of log-likelihood, the model’s performance would probably be closer to the true entropy than the ELBO plotted in Figure 3a). What about increasing the capacity of the used in this paper? This makes me wonder what exactly the R-D plot can teach us about building better generative models.

The toy example in Figure 2 is interesting. What does it tell us about how to build our generative models? Should we be using powerful decoders but a lower beta?

The authors write: “we are able to learn many models that can achieve similar generative performance but make vastly different trade-offs in terms of the usage of the latent variable”. Yet in Figure 3b) it appears that changing the rate of a model can influence the generative performance (ELBO) quite a bit?

---

> ### Author Response · Authors · 2018-01-05
> **Response to AnonReviewer2**
>
> Based on your feedback, we have taken more care to clarify that the core variational bounds we present at the beginning of Section 2 and in some of the appendices were originally derived in Barber and Agakov 2003, Agakov 2006, and Alemi et al. 2017. We have kept the proofs in the appendix for clarity.
>
> We view the core contributions of our work as (1) providing a clear theoretical justification for the beta-VAE objective, and (2) demonstrating that the beta-VAE objective can be used to target a task-specific target rate independent of architecture. This alleviates a known problem with the ELBO objective: achieving different rates requires modifying the architecture and is difficult to control. We have attempted to make those points more clearly in the current version of the paper.
>
> Understanding the origin of the beta-VAE objective opens the door to other approaches for training VAEs. The updated conclusion presents some of these approaches, such as constrained optimization on the variational bounds of rate and distortion, and using more powerful mutual information predictors.
>
> Finally, analyzing the performance of VAE models in terms of the RD plane can make problems with models immediately clear (for example, poor distortion at high rates or poor use of latent variables with complex decoders).  We also hope that we've clearly established some connections between information theory, latent variable models, rate-distortion theory, and compression, which could spawn new results.
>
> > Based on the R-D plots, the authors identify a potential problem of generative models,
> > namely that none of the trained models appear to get close to the “auto-encoding limit”
> > where the distortion is close zero. Wouldn’t this gap easily be closed by a model with identity
> >  encoder, identity decoder, and PixelCNN++ for the marginal distribution?
>
> We agree that models with more powerful PixelCNN++ type marginals could help to close the gap in ELBO for models at high rates. The recent VQ-VAE work shows this qualitatively, and future work should more extensively explore these models to identify their frontier in the RD plane.
>
> > The toy example in Figure 2 is interesting. What does it tell us about how to build our
> > generative models? Should we be using powerful decoders but a lower beta?
>
> The toy example shows that the most powerful models (in this case, models that are able to perfectly represent all relevant distributions) will perform very poorly when trained using ELBO, but constraining the optimization to some rate, either using something like a beta-VAE with beta < 1, or explicitly targeting a known rate, can result in using the model capacity optimally. As seen on MNIST and Omniglot, it seems that the best models we currently have use powerful decoders and powerful marginals, but it is necessary to use beta < 1 to avoid rate collapse.
>
> > “we are able to learn many models that can achieve similar generative performance but
> > make vastly different trade-offs in terms of the usage of the latent variable”.
>
> We meant to speak more directly to the performance over intermediate rate values, from 0 to around 10 nats on MNIST.  In terms of ELBO these models are all the same, but as demonstrated in Figure 4, over these rates we move from an unconditional generative model to an effective compression model for MNIST that seems to preserve the salient features of the input. As we describe elsewhere in the paper, the high-rate models we explored cannot attain equivalently good marginal log likelihood.

---

### Official Review · AnonReviewer1 · 2017-11-27
**Review: Valuable formulation of VAE training tradeoffs**

**Rating:** 7
**Confidence:** 5

**Review:**

EDIT:  I have reviewed the authors revisions and still recommend acceptance.


Summary

This paper proposes assessing VAEs via two quantities: rate R (E[ KLD[q(z|x) || p(z)] ]) and distortion D (E[ log p(x|z) ]), which can be used to bound the mutual information (MI) I(x,z) from above and below respectively (i.e. H[x] - D <= I(x,z) <= R).  This fact then implies the inequality H[x] <= R + D, where H[x] is the entropy of the true data distribution, and allows for the construction of a phase diagram (Figure 1) with R and D on the x and y axis respectively.  Models can be plotted on the diagram to show the degree to which they favor reconstruction (D) or sampling diversity (R).  The paper then reports several experiments, the first being a simulation to show that a VAE trained with vanilla ELBO cannot recover the true rate in even a 1D example.  For the second experiment, 12 models are trained by varying the encoder/decoder strength (CNN vs autoregressive) and prior (fact. Gauss vs autoregressive vs VampPrior).  Plots of the D vs R and ELBO vs R are shown for the models, revealing that the same ELBO value can be decomposed into drastically different R and D values.  The point is further made through qualitative results in Figure 4.


Evaluation

Pros:  While no one facet of the paper is particularly novel (as similar observations and discussion has been made by [1-4]), the paper, as far as I’m aware, is the first to formally decompose the ELBO into the R vs D tradeoff, which is natural.  As someone who works with VAEs, I didn’t find the conclusions surprising, but I imagine the paper would be valuable to someone learning about VAEs.  Moreover, it’s nice to have a clear reference for the unutilized-latent-space-behavior mentioned in various other VAE papers.  The most impressive aspect of the paper is the number of models trained for the empirical investigation.  Placing such varied models (CNN vs autoregressive vs VampPrior etc) onto the same plot from comparison (Figure 3) is a valuable contribution.

Cons:  As mentioned above, I didn’t find the paper conceptually novel, but this isn’t a significant detraction as its value (at least for VAE researchers) is primarily in the experiments (Figure 3).  I do think the paper---as the ‘Discussion and Further Work’ section is only two sentences long---could be improved by providing a better summary of the findings and recommendations moving forward.  Should generative modeling papers be reporting final R and D values in addition to marginal likelihood?  How should an author demonstrate that their method isn’t doing auto-decoding?  The conclusion claims that “[The rate-distortion tradeoff] provides new methods for training VAE-type models which can hopefully advance the state of the art in unsupervised representation learning.”  Is this referring to the constrained optimization problem given in Equation #4?  It seems to me that the optimal R-vs-D tradeoff is application dependant; is this not always true?

Miscellaneous / minor comments:  Figure 3 would be easier to read if the dots better reflected their corresponding tuple (although I realize representing all tuple combinations in terms of color, shape, etc is hard).  I had to keep referring to the legend, losing my place in the scatter plot.  I found sections 1 and 2 rather verbose; I think some text could be cut to make room for more final discussion / recommendations.  For example, I think the first two whole paragraphs could be cut or at least condensed and moved to the related works section, as they are just summarizing research history/trends.  The paper’s purpose clearly starts at the 3rd paragraph (“We are interested in understanding…”).  The references need cleaned-up.  There are several conference publications that are cited via ArXiv instead of the conference (IWAE should be ICLR, Bowman et al. should be CoNLL, Lossy VAE should be ICLR, Stick-Breaking VAE should be ICLR, ADAM should be ICLR, Inv Autoregressive flow should be NIPS, Normalizing Flows should be ICML, etc.), and two different versions of the VAE paper are cited (ArXiv and ICLR).


Conclusions

I found this paper to present valuable analysis of the ELBO objective and how it relates to representation learning in VAEs.  I recommend the paper be accepted, although it could be substantially improved by including more discussion at the end.



1.  S. Zhao, J. Song, and S. Ermon.  “InfoVAE: Information Maximizing Variational Autoencoders.”  ArXiv 2017.

2.  X. Chen, D. Kingma, T. Salimans, Y. Duan, P. Dhariwal, J. Shulman, I. Sutskever, and P. Abbeel.  “Variational Lossy Autoencoder.”  ICLR 2017.

3.  I. Higgins, L. Matthey, A. Pal, C. Burgess, X. Glorot, M. Botvinick, S. Mohamed, and A. Lerchner. “Beta-VAE: Learning Basic Visual Concepts with a Constrained Variational Framework.”  ICLR 2017

4.  S. Bowman, L. Vilnis, O. Vinyas,  A. Dai, R. Jozefowicz, and S. Bengio.  “Generating Sentences from a Continuous Space.”  CoNLL 2016.

---

> ### Author Response · Authors · 2018-01-05
> **Response to AnonReviewer1**
>
> Thanks to your feedback we have substantially improved our References and Discussion section.
>
>  > It seems to me that the optimal R-vs-D tradeoff is application dependant; is this not always true?
>
> We agree, and we have clarified that in the current revision.  The optimal R-D tradeoff is application specific.  Originally we set out on this work and the information theoretic treatment precisely to try to investigate how current VAEs arrived at their particular rates.  When we did our analysis and followed the natural steps to turn it into an easy-to-optimize objective, we discovered that the result was the beta-VAE objective.  We feel that giving a principled motivation for why the beta-VAE objective itself is useful and what it can accomplish is novel.  By better understanding the origin of the beta-VAE objective, we hopefully open the door to more and better work on theoretically-motivated objective functions in the future.

---

### Official Review · AnonReviewer3 · 2017-11-28
**Good analysis. Contributions are not so clear though.**

**Rating:** 5
**Confidence:** 5

**Review:**


- I think that VAEs are rather forced to be interpreted from an information theoretic point of view for the sake of it, rather than for the sake of a clear and unequivocal contribution from the perspective of VAEs and latent-variable models themselves. How is that useful for a VAE?

- "The left vertical line corresponds to the zero rate setting. ...": All these limits are again from an information theoretic point of view and no formulation nor demonstration is provided on how this can actually be as useful. As mentioned earlier in the paper, there are well-known problems with taking this information theory perspective, e.g. difficulties in estimating MI values, etc.

- Breaking (some of) the long sentences and paragraphs in page 3 with an unequivocal mathematical formulation would smooth the flow a bit.

- "(2) an upper bound that measures the rate, or how costly it is to transmit information about the latent variable.": I am not entirely sure about this one and why it is massively important to be compromised against the obviously big first term.

- Toy Model experiment: I do not see any indication of how this is not just a lucky catch and that VAEs consistently suffer from a problem leading to such effect.

- Section 5: "can shed light on many different models and objectives that have been proposed in the literature... ": Again the contribution aspect is not so clear through the word "shed light".


Minor
- Although apparently VAEs represent the main and most influential latent-variable model example, I think switching too much between citing them as VAEs and then as latent-variable models in general was a bit confusing. I propose mentioning in the beginning (as happened) that VAEs are the seminal example of latent-variable models and then going on from this point onwards with VAEs without too much alternation between latent-variable models and VAEs.

- page 8: "as show"

---

> ### Author Response · Authors · 2018-01-05
> **Response to AnonReviewer3**
>
> > I think that VAEs are rather forced to be interpreted from an information theoretic point
> > of view for the sake of it, rather than for the sake of a clear and unequivocal contribution
> > from the perspective of VAEs and latent-variable models themselves. How is that useful for a
> > VAE?
>
> Our analysis explains why VAEs with strong decoders can learn to ignore the learned latent variables. The analysis directly leads to a derivation of the beta-VAE objective, which can force any particular VAE architectural choice to make appropriate use of the latent variables. The value to practitioners using VAEs is to encourage them to use the beta-VAE objective to overcome this shortcoming of standard ELBO optimization.
>
> We believe the information theoretic perspective gives a natural motivation for the beta-VAE objective, not just as a simple modification of the objective with observed effect, but demonstrates that it allows you to explore the entire rate-distortion frontier for a particular model family.  This is useful and necessary since the relative power of the encoder / decoder and marginal are hard to tune. As  observed in the current literature, powerful autoregressive decoders tend to collapse to vanishing rate at beta=1.
>
> > "The left vertical line corresponds to the zero rate setting. ...": All these limits are again from
> > an information theoretic point of view and no formulation nor demonstration is provided on
> >  how this can actually be as useful. All these limits are again from an information theoretic
> > point of view and no formulation nor demonstration is provided on how this can actually be
> > as useful.
>
> and
>
> > "(2) an upper bound that measures the rate, or how costly it is to transmit information about
> > the latent variable.": I am not entirely sure about this one and why it is massively important
> > to be compromised against the obviously big first term.
>
> We hope that the modifications to the paper make these points more clear. We think that our experiments convincingly demonstrate that low rates and high rates give qualitatively different model behavior, as seen in Figures 4 and 6, and it is exactly the tradeoff between rate and distortion that produces that different behavior, since the models are otherwise held constant. Very low rate models fail at reconstruction, as the information theory predicts. Low rate models manage to capture the semantics of the digits, but ignore the style in reconstruction, and higher rate models provide more precise reconstructions, but fail to provide variation during generation.
>
> > Toy Model experiment: I do not see any indication of how this is not just a lucky catch and
> > that VAEs consistently suffer from a problem leading to such effect.
>
> We have added some text indicating that both the VAE and target rate results were stable across all of the random initializations we performed (many dozens). The core point of the toy model was to illustrate that the normal ELBO (beta=1 VAE) objective does not target any rate in particular.  The rate it ends up at is a complicated function of the relative powers of the component models. For a given problem, we as practitioners usually have some inductive bias for how much information we believe is relevant and should be retained; by targeting the rate we knew to be relevant in the toy model, we were able to nearly perfectly invert the true generative model.  We believe our experiments on both MNIST and Omniglot further demonstrate that for most model architectures, the rates achieved optimizing ELBO with powerful models are often low.
>
> > - Section 5: "can shed light on many different models and objectives that have been
> > proposed in the literature... ": Again the contribution aspect is not so clear through the word
> > "shed light".
>
> Our discussion section was lacking.  We've expanded it, thank you.

---

### Public Comment · ~Ethan_Fetaya4 · 2017-11-16
**Great work, overly complex proof of B.3 and B.4**

Really enjoyed your paper, gave very useful insights.

One small thing, the proofs for the optimal encoder/decoder are much more complicated then they need be. The only inequality is from the KL divergence "positive semidefinite quality", so the bound is tight exactly when the KL divergence is zero i.e. when the probabilities (a.s.) match and that is all you need.

---

> ### Author Response · Authors · 2018-01-05
> **Thanks**
>
> Thanks for the compliments and the feedback on the proofs.  We've left in the detailed derivations in the Appendix for completeness and clarity.

---

### Author Response · Authors · 2018-01-05
**General Response to Reviewers**

We thank all the reviewers for their valuable feedback. We have made a number of improvements and clarifications that we believe amount to a substantially improved version of the paper. Those changes are summarized and discussed below.

The main concern from all three reviewers was that of novelty and the “forced” perspective of the ELBO in terms of information theory. To address these issues, we have extended the discussion and introduction sections to highlight the utility of this perspective. In summary, we view the primary contributions of our paper to be the following, and we think the new version improves on the presentation of most of these points. We have:
 * Clarified our core point, which is that the best rate is task-dependent, and if you only optimize the ELBO the architecture determines the rate.
 * Motivated the beta VAE objective as a natural objective for exploring the entire frontier of rate-distortion tradeoffs for a model.
 * Demonstrated the use of rate-distortion diagrams to visualize tradeoffs in latent variable models.
 * Motivated reporting of rate and distortion in future papers on generative models since different applications require different tradeoffs which cannot be extracted from the ELBO alone.
* Provided a simpler explanation of and solution to the problem of powerful decoders ignoring the latent variables than the explanations and solutions previously proposed in the literature (e.g., Chen et al., Variational Lossy Autoencoder, 2017).
 * Comprehensively explored the relative performance and tradeoffs of a wide range of current modelling choices on MNIST and Omniglot

Additionally, we:
 * Fixed the issues noted, including a substantial cleanup of the References.
 * Added many more Omniglot experimental results in Appendix A.

As highlighted by R1, our paper is the first to extensively evaluate the tradeoffs made by different architecture choices by varying the complexity of the posterior, prior, and decoder. Many recent papers have been published on subsets of these architecture choices, for example more complex priors (VampPrior), latent variable models with autoregressive decoders (PixelGAN, PixelVAE), or more complex posteriors (IAF, normalizing flows). Our experiments highlight that different models dominate in different regimes, in particular autoregressive decoders are critical at low rates while VampPrior works well at intermediate and high rates.

Finally, as suggested by R2, we have added a call for researchers to report rate and distortion separately in future publications. In particular, this highlights the tradeoffs being made by new approaches and points to models that are learning useful representations (with non-zero rate). For practitioners interested in learning representations, our paper highlights the need to look beyond the ELBO as there exists a continuum of models with the same ELBO but dramatically different rates.

These changes represent a substantial improvement in the paper, and we hope the reviewers will take these into consideration.

---

### Decision · Program_Chairs · 2018-01-29
**ICLR 2018 Conference Acceptance Decision**

**Decision:**

Reject

**Comment:**

This paper gives a coding theory interpretation of VAEs and uses it to motivate an additional knob for tuning and evaluating VAEs: namely, the tradeoff between the rate and the distortion. This is a useful set of dimensions to investigate, and past work on variational models has often found it advantageous to penalize the latent variable and observation coding terms differently, for broadly similar motivations. This paper includes some careful experiments analyzing this tradeoff for various VAE formulations, and provides some interesting visualizations. However, as the reviewers point out, it's difficult to point to a single clear contribution here, as the coding theory view of variational inference is well established, and the VAE case has been discussed in various other works. Therefore, I recommend rejection.